**Subject Category:**
Biology (whole organism)

evolution

animalia, morphology, ultrastructure, large molecules

**Author for correspondence:**
Claus Nielsen
e-mail: cnielsen@snm.ku.dk

# Early animal evolution: a morphologist's view

## Claus Nielsen

The Natural History Museum of Denmark, University of Copenhagen, Zoological Museum, Universitetsparken 15, DK-2100 Copenhagen, Denmark

CN, 0000-0001-6898-7655

Two hypotheses for the early radiation of the metazoans are vividly discussed in recent phylogenomic studies, the 'Porifera-first' hypothesis, which places the poriferans as the sister group of all other metazoans, and the 'Ctenophora-first' hypothesis, which places the ctenophores as the sister group to all other metazoans. It has been suggested that an analysis of morphological characters (including specific molecules) could throw additional light on the controversy, and this is the aim of this paper. Both hypotheses imply independent evolution of nervous systems in Planulozoa and Ctenophora. The Porifera-first hypothesis implies no homoplasies or losses of major characters. The Ctenophora-first hypothesis shows no important synapomorphies of Porifera, Planulozoa and Placozoa. It implies either independent evolution, in Planulozoa and Ctenophora, of a new digestive system with a gut with extracellular digestion, which enables feeding on larger organisms, or the subsequent loss of this new gut in the Poriferans (and the re-evolution of the collar complex). The major losses implied in the Ctenophora-first theory show absolutely no adaptational advantages. Thus, morphology gives very strong support for the Porifera-first hypothesis.

## 1. Introduction

Until recently, the poriferans have almost unanimously been regarded as the sister group of all other animals. Haeckel [1] even placed them with the 'protists', outside the animals. However, the position of the poriferans as the sister group of the eumetazoans became challenged when large phylogenomic studies including two ctenophore genomes showed the Ctenophora as the sister group to all the other metazoans [2,3]. Several later phylogenomic studies have found support for this 'Ctenophora-first' theory [4,5]. However, another set of large phylogenomic studies favour the alternative 'Porifera-first' theory [6–10].

Some recent phylogenomic studies simply conclude that the position of deep-diverging lineages, such as the ctenophores, may exceed the limit of resolution afforded by the traditional phylogenomic analyses [4,10,11]. It therefore appears timely to

**Table 1.** Occurrence of a selection of morphological and molecular characters in choanoflagellates and the main metazoan phyla. Collar complexes consist of a ring of microvilli surrounding an undulating cilium and function in water transport and particle collection. White dots—absent; red dots—present; characters in parenthesis are not found in all species. Blue—characters of the Metazoa; green—characters of the Eumetazoa; orange—characters of the Parahoxozoa; yellow—characters of the Cnidaria (+Bilateria, i.e. Planulozoa); grey—characters of the Ctenophora.

| characters | Choanoflagellata | Porifera | Cnidaria | Placozoa | Ctenophora | advanced choanoblastaea | gastraea | major references |
|---|---|---|---|---|---|---|---|---|
| mesoderm with myocytes | ○ | ○ | ○ | ○ | ● | ○ | ○ | Ryan *et al.* [2] |
| colloblasts | ○ | ○ | ○ | ○ | ● | ○ | ○ | Franc [17] |
| neurotransmitters B | ○ | ○ | ○ | ○ | ● | ○ | ○ | Moroz & Kohn [18] |
| aboral organ | ○ | ○ | ○ | ○ | ● | ○ | ○ | Norekian & Moroz [19] |
| neurotransmitters A | ○ | ○ | ● | ○ | ○ | ○ | ○ | Moroz & Kohn [18] |
| apical organ | ○ | ○ | ● | ○ | ○ | ○ | ○ | Marlow *et al.* [20] |
| cnidae | ○ | ○ | ● | ○ | ○ | ○ | ○ | David *et al.*, [21] |
| nidogen, perlecan | ○ | ○ | ● | ● | ○ | ○ | ○ | Dayraud *et al.* [22], Fidler *et al.* [23] |
| HIF respiratory pathway | ○ | ○ | ● | ● | ○ | ○ | ○ | Mills *et al.* [24] |
| eumetazoan genes | ○ | ○ | ● | ● | ● | ○ | ○ | Srivastava [25] |
| extracellular digestion | ○ | ○ | ● | ● | ● | ○ | ● | Smith *et al.* [26] |
| ectoderm+endoderm | ○ | ○ | ● | ● | ● | ○ | ● | Hashimshony *et al.* [27] |
| metazoan genes | ○ | ● | ● | ● | ● | ● | ● | Srivastava *et al.* [28] |
| collagen IV | ○ | ● | ● | ○ | ● | ● | ● | Fidler *et al.* [23] |
| basement membrane | ○ | (●) | ● | ● | (●) | ● | ● | Fidler *et al.* [23] |
| epithelia | ○ | ● | ● | ● | ● | ● | ● | Leys *et al.* [29], Fidler *et al.* [23] |
| oogamy, diploidy | ○ | ● | ● | ● | ● | ● | ● | Levin *et al.* [30], Woznica *et al.* [31] |
| multicellularity | ○ | ● | ● | ● | ● | ● | ● | |
| intracellular digestion | ● | ● | ○ | ○ | ○ | ● | ○ | Simpson [32], Vacelet & Duport [33] |
| cells with collar complex | ● | ● | ○ | ○ | ○ | ● | ○ | Mah *et al.* [34], Laundon *et al.* [16] |

see if a phylogenetic hypothesis can be constructed based exclusively on morphology in the widest sense, i.e. including molecules, for comparison with the phylogenomic hypotheses.

The review is almost exclusively based on papers which contain information about the Choanoflagellata and the four major metazoan groups, Porifera, Cnidaria (as a representative of the Planulozoa, i.e. Cnidaria + Bilateria), Placozoa and Ctenophora.

# 2. The sister group of the Metazoa: Choanoflagellata

Almost all recent authors regard the choanoflagellates as the sister group of the metazoans [12]. The closest outgroups, ichthyosporeans and filastraeans are parasites or commensals, with life cycles comprising uninucleate, opisthokont swarmers and colonial or plasmodial adults [13]. They may all be haploid [14]. The ichthyosporean *Sphaeroforma arctica* has a multinucleate stage, and cellularization leads to the formation of a polarized layer of individual cells resembling an epithelium without a basement membrane [15].

The choanoflagellates are solitary or colonial organisms with a 'collar complex' consisting of an undulating cilium (usually called flagellum), with a vane, surrounded by a conical collar of microvilli [12,16] (table 1). Unicellularity precludes differentiation into non-feeding cell types, because there is no exchange of nutrients between the cells. The cilium propels water away from the cell body (opisthokont), and microscopic particles, mostly bacteria, are caught at the outside of the collar and ingested by pseudopodia. Digestion is poorly known, but it is believed that the particles disintegrate inside a vacuole and the molecules become absorbed. Sexual reproduction seems to be rare, but isogamy (or slightly different sizes of otherwise similar gametes) has been observed in cultures of *Salpingoeca rosetta* [30,31]. The cells are haploid, but a short diploid phase after fertilization may occur [30].

Large numbers of genes involved in organizing structures in metazoans, such as epithelia and nervous systems, are present already in choanoflagellates, where their functions are unknown

[13,28,35]. Obviously, the presence of such genes, as, for example, genes of the postsynaptic scaffold found in choanoflagellates and poriferans [11,36], does not indicate that these organisms are descendants of organisms with a nervous system!

# 3. Origin of the Metazoa (Animalia): the latest common ancestor, choanoblastaea

It is generally hypothesized that the sponges evolved from clonal colonies of choanoflagellate-like ancestors [13], perhaps via a 'rosetta' stage [34].

Choanoflagellates and choanocytes are generally very similar, most differences probably being related to their different positions, choanoflagellates being single cells or cells in small colonies, whereas the sponge choanocytes are arranged in choanocyte chambers situated in interior water canals [16,34].

Evolution of the ancestral metazoan, choanoblastaea, probably passed from a sphere of clonal, feeding choanoflagellates to the multicellular ancestor [37,38]. The establishment of an epithelium with tight cell contacts [29] enabled the exchange of nutrients between the cells, and this made the evolution of non-feeding cells possible, both cells at the surface of the sphere and internal cells. Many types of ciliated eumetazoan cells show their evolution from choanoblastaea cells, having a ring of shorter or longer stiff microvilli around a non-contractile cilium, for example, many types of sensory cells and the weir cells in protonephridia, but their function is never particle collecting [38]; these cells have been called choanocytes but are better-called collar cells. The sharing of nutrients paved the way to other metazoan specializations, such as oogamy and diploidy [39]. A stabilizing basement membrane probably evolved at an early point. The associated collagen IV is present in all the metazoan groups but absent in the demosponge *Amphimedon* [23]. The occurrence of the basement membrane is scattered in the non-bilaterian animals, with presence or absence in sponges and ctenophores, absence in *Trichoplax* and presence in cnidarians and in almost all bilaterians (table 1).

There is an impressive addition of novel genes and gene families at the base of the Metazoa [40], for example, cell–cell adhesion by β-catenin regulated cadherins and developmental signalling pathways, such as WNT and transforming growth factor-β [13,25,40–42].

# 4. Origin and evolution of the Porifera

There are different theories for the origin of the sponges [43] but only the theory of Nielsen [37] presents a continuous series of evolutionary stages based on changes in structure and function. This theory explains how an advanced choanoblastaea settled by an area of unciliated cells and organized the choanocytes in a groove, and how this groove became transformed into a choanocyte chamber. This was the ancestral sponge, which with increasing body size developed more choanocyte chambers organized in an aquiferous system.

Digestion is intracellular [32,33], as that of the choanoflagellates, with particles in vacuoles disintegrating, apparently without being digested by enzymes [44]. One clade of demosponges [45] has lost the collar chambers and specialized in feeding on small captured animals, such as crustaceans; the digestion is again intracellular without secreted enzymes [33].

Sponges have epithelia with various types of cell junctions [23,29].

Muscle cells do not occur in the sponges, but the whole body and, for example, the ostia of demosponges can change the shape by contraction of the pinacoderm, and small specimens can even crawl on the substratum [46,47]. The 'myocytes' (actinocytes) of the mesohyle have been reported to be contractile, but this is uncertain [48].

The sponges lack a nervous system, but demosponge behaviour indicates communication between cells [49,50]. Cells at the oscula with a short, stiff cilium are supposed to be sensory [51]. In the hexactinellids, ciliary activity is controlled by electrical impulses in the whole syncytium [52]. The lack of a nervous system has been interpreted as a loss [53], but the explanation is highly imaginative!

Some early steroid sponge-markers are from the Ediacaran, *ca* 660–635 Ma [54], but this has been questioned [55]. The earliest reasonably certain sponge spicules are from the earliest Cambrian, about 540 Ma [56]. This has been characterized as a 'fossil gap' [57], but it should be remembered that the defining character of a sponge is the presence of an aquiferous system with choanocyte chambers, not the presence of a skeleton, so the poriferan ancestor must have preceded an ancestral 'thin-walled, hexactine-based sponge' [56]. It is much more probable that the earliest sponges were small and without a skeleton, and that they therefore have left no body fossils.

# 5. Origin of the Eumetazoa: the latest common ancestor: gastraea

The eumetazoans comprise three main groups, Planulozoa (Cnidaria + Bilateria, in the following discussion represented by the Cnidaria), Placozoa and Ctenophora. The most obvious character, which distinguishes the eumetazoans from the sponges is the division of the epithelium into an inner or 'ventral' layer of digestive cells, endoderm, and an outer or 'dorsal' protective ectoderm. This was probably followed by a loss of digestive functions in the ectodermal cells [27]. The digestive epithelium is invaginated as an archenteron in planulozoans and ctenophores but forms a creeping sole in the placozoans. Digestion is extracellular with digestive enzymes secreted into the archenteron (or in placozoans into a temporarily sealed space between the endoderm and the substratum), and digested materials become absorbed by the endodermal cells. Unfortunately, there seems to be no study of the digestive enzymes of placozoans and ctenophores, which could provide important phylogenetic information. The new organization of feeding and digestion opens up a wealth of new lifestyles, and the morphological diversity of the eumetazoans is enormous compared to that of the sessile poriferans, which are limited to feeding on bacteria and small particles. The Ctenophora-first theory implies either that the poriferans lost the gut and reverted to intracellular digestion or that gut and extracellular digestion evolved convergently in ctenophores and parahoxozoans; both possibilities appear highly improbable.

Movements were mainly by cilia on monociliate cells, but changes in shape may have been possible through contractions of actin filaments, as seen in *Trichoplax* (see below). A nervous system and muscles were not present, but the cells could have had peptidergic communication, as seen in *Trichoplax* [58].

A large number of genes found in the eumetazoans are absent from the poriferans [41], and a number is absent from the ctenophores too [6].

# 6. Evolution of the Eumetazoa: the relationship between Cnidaria and Ctenophora

The classical Coelenterata hypothesis was based mainly on general morphological similarities between the pelagic medusa and the (usually) pelagic ctenophores, but the ancestral cnidarian was most likely a polyp [59]. The two groups share a general body organization with ectoderm and invaginated digestive endoderm (characteristic of the gastraea), but when studied more closely, the differences between the two groups become very apparent, indicating separate evolution from the gastraea [60].

The cnidarian larvae have an apical organ with a tuft of cilia, like that of many ciliated bilaterian larvae; this organ is probably a chemoreceptor related to larval metamorphosis [20]. The ctenophores lack a primary larva and hatch as small juveniles, which have an obviously non-homologous aboral organ (retained in the adults) with a dome of cilia covering four compound balancer cilia supporting a ball of statoliths, i.e. a gravity-sensing organ [19,61,62].

The cnidarians have cnidae, cells containing a collagen-like sphere with an invaginated tube, which can be everted and inject various toxins into prey or predator [21], and the ctenophores have colloblasts, mushroom-shaped cells with numerous vesicles with adhesive substances [17]. The cnidae found in the ctenophore *Haeckelia* originate from ingested medusae [63].

The cnidarian gut is a sac with only one opening, functioning as both mouth and anus, whereas the ctenophores have a gut with a mouth, a stomach and a pair of anal canals leading to temporal anal openings (pores) [19,64,65].

Cnidarians have myoepithelial cells (and the bilaterians additionally have mesodermal muscles), whereas the ctenophores have both an epithelial net of muscles and mesogloeal myocytes [19,22,61]. The majority of signalling molecules and transcription factors involved in specifying and differentiating the mesoderm of bilaterian animals are absent in the ctenophore *Mnemiopsis* [2]. Striated muscle cells are found in most bilaterians, but the scattered occurrence in the tentillae of the ctenophore *Euplokamis* and in the hypocodon of hydrozoan medusae [66] must be regarded as homoplasies [5].

The ability to communicate fast and precisely between cells/organs in the more active eumetazoans is necessary for their more active, often vagile lifestyles. This is made possible through the evolution of nervous systems. These systems consist of neurons, i.e. cells dedicated to transmitting information from one cell to another, usually through electric impulses. A neuron consists of a cell body with the nucleus (perikaryon) usually with a long, thin neurite; bundles of neurites are called nerves [67,68]. Connections between the cells are of two types, chemical (synapses) and electrical (gap junctions). Cnidarians and ctenophores have nervous systems without a brain, whereas the bilaterians have a

ventral nervous system and an anterior, dorsal brain [69]. The prevailing theory for the evolution of the nervous system appears to be that of Mackie [70], which proposes that neurons and myocytes differentiated from myoepithelial cells. This could well have taken place more than once, as suggested by a number of studies [3,71]. However, other theories have been proposed [72].

The synapses of cnidarians (and bilaterians) generally have the 'usual' ultrastructure with a simple arrangement of synaptic vesicles along the contact with the effector cell [73], whereas the ctenophore synapses show a specialized 'triad' structure with a series of vesicles and a thin extension of the endoplasmatic reticulum along the cell contacts [74,75]. The isolated occurrence of bidirectional, triad-like synapses in the medusa of the scyphozoan *Cyanea* [76,77] is most likely a homoplasy.

There are major differences between the neurotransmitters in the cnidarian-bilaterian synapses, which use, e.g. acetylcholine, glutamate and GABA (here called neurotransmitters A), and the ctenophores, with mainly peptidergic synapses (here called neurotransmitters B) [18]; the two groups do not share any of the specific neurotransmitters [74]. This is a strong indication of a separate evolution of synapses in cnidarians and ctenophores. Also, the neuronal-specific genes of the two groups are completely different, with the ctenophores lacking, for example, neurogenins, NeuroD, Achaete-scute, REST, HOX and Otx [3,71].

Invertebrate gap junctions consist of a ring of eight innexin units, surrounding a pore which can be opened and closed [78]. These pores permit the exchange of small molecules between the cells and transmission of electrical impulses [79]. This type of cell junctions is present in cnidarians (and bilaterians) and ctenophores, but although there is a considerable variation in the innexins of planulozoans and ctenophores, the high number of special innexis in the ctenophores could indicate independent evolution [71].

Important components of the HIF respiratory pathway are only found in placozoans, cnidarians and bilaterians [24].

Based on these characters, it appears probable that Cnidaria (+ Placozoa, see below) and Ctenophora are sister groups, which evolved independently from the advanced gastraea. This implies that both nervous system and mesoderm evolved convergently in the two groups.

# 7. Origin of the Parahoxozoa, i.e. Placozoa + Cnidaria (+Bilateria)

It is difficult to identify distinguishing characters of this group, which, however, seems to be found in the majority of the phylogenomic analyses [6,8,80–82]. Most of these analyses resolve the interrelationships of the three groups as Placozoa + (Cnidaria + Bilateria), but the topology (Placozoa + Cnidaria) + Bilateria has been found in some recent studies [9]. The group has been named after the Parahox genes, which are found in all planulozoans (Cnidaria + Bilateria). A Parahox-like gene, *Trox-2*, is found in *Trichoplax*, although the authors note that this could be either a true ParaHox gene (*Gsx*) or a 'ProtoHox' gene diverging before distinct Hox and ParaHox genes arose [83]. Either possibility is consistent with a clade comprising Placozoa, Cnidaria and Bilateria. The interpretation of this as a synapomorphy uniting these three groups was challenged by the discovery of a putative Parahox gene (*Cdx*) in calcareous sponges [84]; however, this is still being discussed [85].

The basement membrane contains nidogen and perlecan, which are restricted to the Parahoxozoa [23]. Also, the Zic genes seem to be restricted to placozoans and planulozoans; they are expressed at the lateral edges of the neural plate in nematodes, annelids and chordates [86]. Only the parahoxozoans have the complete HIF respiratory pathway [24].

# 8. Morphology and position of the Placozoa

The flattened, amoeboid placozoans consist of a pseudostratified 'ventral' epithelium, which serves as a digestive creeping sole, consisting mainly of monociliate cells with long microvilli, and a 'dorsal' ectoderm consisting of thin monociliate cells. The 'dorsal–ventral' orientation is supported by studies of gene expression [87]. Between these layers are found a number of fibre cells with long, branching processes, which contact all epithelial cells without forming organized junctions [26,88]. The 'amoeboid' movements of the organism appear to be caused by the contraction of actin filaments in the fibre cells [89,90]. There is no nervous system, but peptidergic signalling between the various cells induces various types of behaviour [58], and a ring of gland cells at the periphery are FRMFamide immunoreactive [91]. The epithelial cells are united by adherens junctions with apical actin networks [92], and contractions of this network may contribute to changes in the shape of the body [93]. Cells

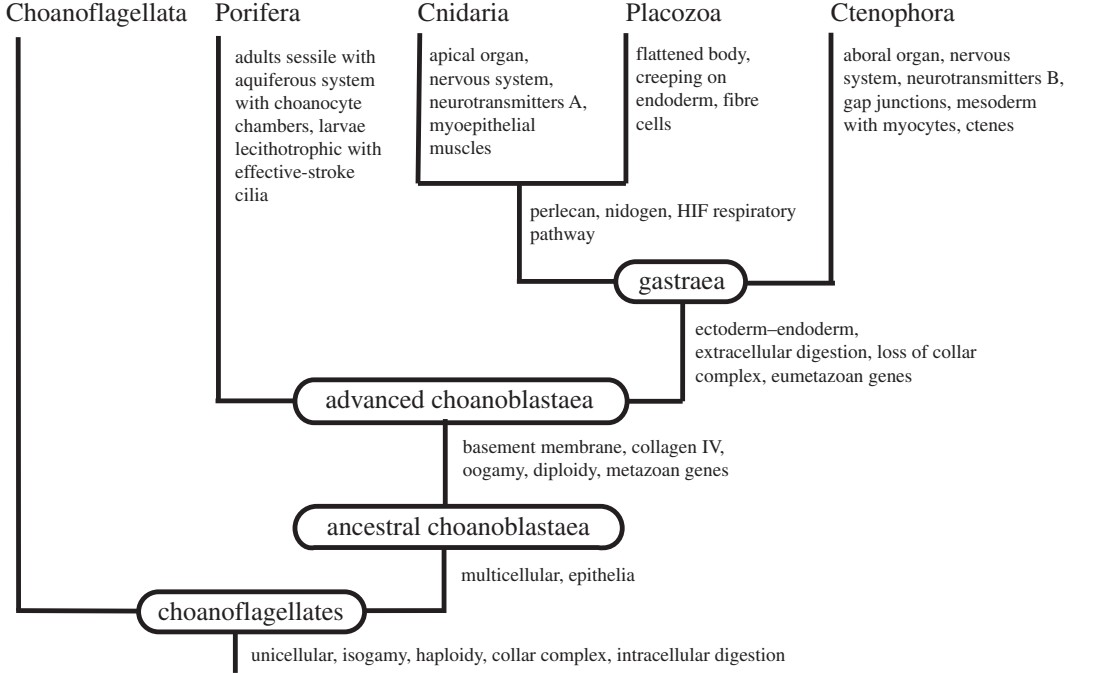

**Figure 1.** Early animal phylogeny according to the Porifera hypothesis with an indication of gains of important characters. Neurotransmitters A and B, see the text.

in the lower epithelium secrete digestive enzymes into the sealed space between the epithelium and the substratum, and digested material becomes absorbed [26].

For many years, only one placozoan species, *Trichoplax adherens*, was recognized, but molecular studies have revealed a considerable diversity of species [94], and new genera are now being described [95].

The Ediacaran fossil *Dickinsonia* (571–541 Ma) could perhaps best be interpreted as an upper, protective sheet of a placozoan (see also [96]).

The interpretation of *Trichoplax* as a descendant of the gastraea implies that the gastraean ancestor expanded the periphery of the body with the ciliated digestive endoderm becoming the creeping digestive sole. At first, this appears less plausible, but a similar lifestyle is actually seen in the polyp stage of the hydrozoan *Polypodium*, which lives parasitic in eggs of sturgeons. Inside the sturgeon egg, the zygote develops into a planula-like larva, but with an outer layer of monociliated digestive cells, which later becomes the endoderm, surrounding an inner layer, which becomes the ectoderm. An inverted stolon with polypide buds develops inside the endoderm, and when released, the 'larva' evaginates the stolon with the polyps, which begin to feed and produce small medusae [97].

The 'opposite' theory, the plakula theory, proposes that the placozoans represent the ancestral eumetazoan type [98]. This is purely speculative, and a flat plakula stage has never been observed in the ontogeny of any eumetazoan, whereas a gastrula stage has been observed in the ontogeny of almost all eumetazoan phyla [69].

## 9. Discussion

When the major characters discussed above are plotted onto the phylogenetic branching patterns of the Porifera-first and Ctenophora-first hypotheses, a clear picture emerges. Both hypotheses imply that the nervous systems have evolved independently in planulozoans and ctenophorans, so this character has been omitted in the trees.

The Porifera-first hypothesis (figure 1) shows a number of apomorphies in each lineage and no symplesiomorphies or losses (except the loss of the choanocyte collar in connection with the transition to the new mode of feeding/digestion). It resembles 'traditional' morphology-based phylogenies but places the Ctenophora as the sister group of the Parahoxozoa (Placozoa + Planulozoa). At first, this position appears counterintuitive, but it is strongly supported both by most of the phylogenomic studies and by characters from morphology, as shown above.

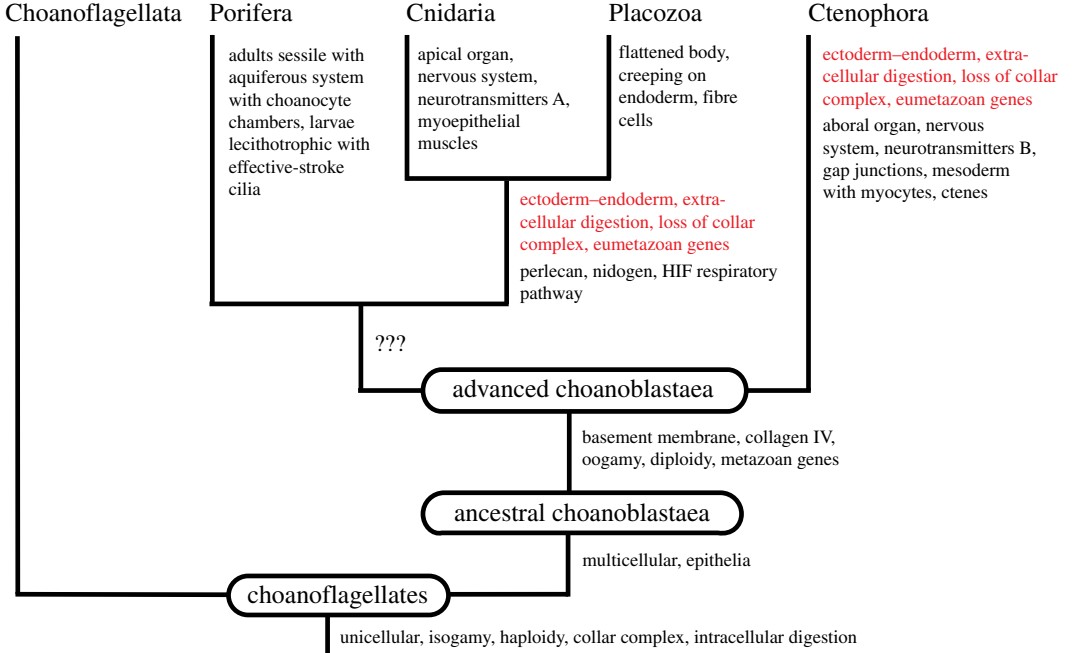

**Figure 2.** Early animal evolution according to the Ctenophora hypothesis (based on homoplasies), with an indication of gains of important characters. Homoplasies are in red.

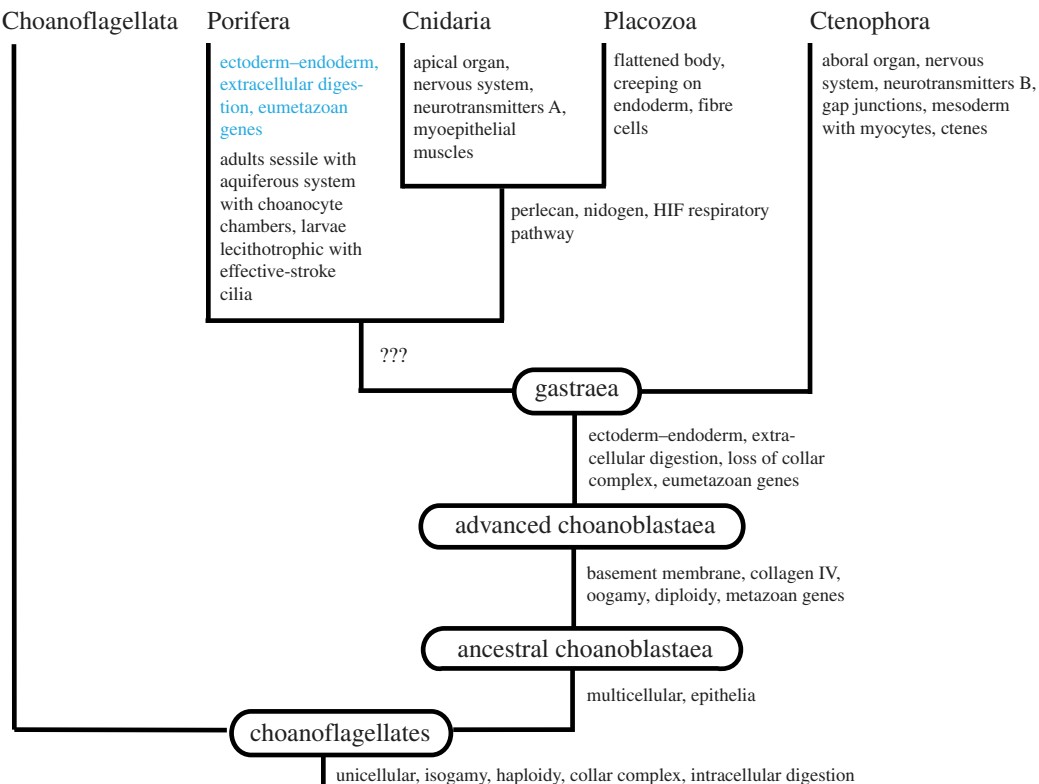

**Figure 3.** Early animal evolution according to the Ctenophora hypothesis (based on losses), with an indication of losses of important characters. Losses are in blue.

The branching pattern of the Ctenophora-first hypothesis implies either a tree based on homoplasies (figure 2) or a tree based on losses (figure 3). Both trees show a lineage of Porifera + Cnidaria + Placozoa without any synapomorphies. The first tree (figure 2) shows that the new digestive system (gut) consisting of a digestive endoderm and extracellular digestion should have evolved independently in Ctenophora + Placozoa and Ctenophora from an advanced choanoblastaea without differentiation of

ectoderm and endoderm and with intracellular digestion. This appears quite unlikely, but studies on the almost unknown digestion of placozoans and ctenophores could throw light on the homology question. The second tree (figure 3) shows that the Porifera should have evolved from a gastraea through the loss of the gut (as described above) and the re-evolution of the collar complex. It implies the loss of the ability to ingest and digest larger organisms, and it is very difficult to see any evolutionary advantage of such a loss. Furthermore, a number of characteristic eumetazoan genes should have been lost.

It must be concluded that the Porifera-first hypothesis finds full support from morphology, whereas the Ctenophora hypothesis implies groups with no synapomorphies and various evolutionary changes which appears most unlikely.

Data accessibility. This article has no additional data.

Competing interests. The author declares that he has no competing interests.

Funding. I received no funding for this study.

Acknowledgements. The author is much indebted to Dr Peter Holland (University of Oxford) for constructive comments to an earlier version of the manuscript. The input from two anonymous reviewers is greatly appreciated.

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
