## [Reviewer comments · Royal Society Open Science]

Review History

RSOS-190638.R0 (Original submission)

Review form: Reviewer 1

Is the manuscript scientifically sound in its present form?

Yes

Are the interpretations and conclusions justified by the results?

No

Is the language acceptable?

Yes

Is it clear how to access all supporting data?

Not Applicable

Do you have any ethical concerns with this paper?

No

Have you any concerns about statistical analyses in this paper?

No

Recommendation?

Major revision is needed (please make suggestions in comments)

Comments to the Author(s)

Nielsen, Early Animal Evolution: A morphologist's view.

Nielsen reviews the morphological characters of each of the four non-bilaterian groups as well as choanoflagellates, with a view to evaluate evolutionary relationships.

The manuscript is a concise summary of the principal features of each group and as such could be not only extremely useful to a newcomer to the field, but also an excellent opinion piece, with a few organizational changes and some additions.

The abstract proposes an interesting hypothesis and although the text presents data, there is very limited analysis of the data, and a simple conclusion at the end saying the data supports the hypothesis is not easily digested. Below are some suggestions for changes that would improve this manuscript.

1. The manuscript sits between review and opinion piece. It would be more useful if it were more objective and if the text really evaluated character gain or loss in the two scenarios proposed, equally: by studying character gain and loss in the case Porifera branched first or in the case Ctenophora branched first. If that were added it would greatly improve the manuscript and increase interest.

2. The headers of subsections outline the author's argument, but given this is about morphology, it seems that the argument would be better placed in a section of its own at the beginning (using those headers) and then new headers might better be titles that highlight key characters that help determine, in a morphologist's view, the likelihood of moving from one state to another (following that argument). Header titles might be for example: 'Unicell to metazoan: cell differentiation' and 'Gaining an epithelium' or 'Epithelia and digestion: the gut'...something along those lines.

For example, choanoflagellates are described as single uniform unicells and the need for phagocytosis by each cell and lack of transfer of material between cells is a character that distinguishes them from metazoans. Some work suggest even similar cells can appear to have distinct characters in colonial flagellates (e.g. Laundon et al 2019), from a morphological view, but this is not yet supported by transcription of molecules. Whether colonies pass materials between one-another has not properly been addressed, and seems the point where Nielsen draws the line. However the text does not remind readers that these organisms are endpoints in evolutionary experiments and so some intermediate form that might have shared nutrients could have existed. Instead the argument is made that collared cells are innately similar, and by parsimony sponges arose from a collared ancestor. This is not necessarily the case however, since not all poriferans have collared cells - collared cells can be lost and gained even within the Porifera.

More emphasis should fall on examining the morphological transition from Porifera through other non-bilaterians and in turn, from Ctenophora through other non-bilaterians including Porifera. An examination of what losses must be considered were Porifera sister to Cnidaria and Placozoa would be helpful. A figure showing the gains on one scenario and losses on the other would be useful.

3. Key features of the different transitions are discussed (e.g. collagens) but not comprehensively. a) For example, Type IV collagen may not in fact be necessary for making epithelia. Even colonial

filisterians can make good epithelia (see Dudin et al on *Sphaeroforma antarctica* in BioArchive), as can slime molds. What glues the cells together in metazoans (collagen) is not one of the characters Nielsen addresses, but seems like it might be a very useful character to evaluate in depth. It should at least be touched on.

b) Phagocytosis and digestion and transfer of food are discussed in one section but needs more attention. Contrary to what is said under the Choanoblastea section, digestion is quite well known in sponges (e.g. Willenz and Van de Vyver, Imsieke, Wilkinson). Can the present ctenophore groups tell us anything about that transition though? What about groups that may not have fossilized, just as Nielsen says that sponges without skeletons would not have fossilized. Absence of fossils means many groups with these characters could have existed. How a gut evolved is addressed under the gastrea section too. It would be useful if one section addressed feeding, digestion and the gut, with all groups.

c) Whether or not nerves arose and were lost is also addressed, but it seems cursory. This topic has been covered heavily by others but the relevance of the gain of nerves as a morphological character could be better discussed - a morphological view on this is lacking in the literature.

4. There are small but significant misunderstandings in these sections. For example, Pg 3 lines 8-9 refer to muscle - but a definition of muscle is needed. Presumably sponges have a type of smooth muscle (most authors from the earliest to recent) find this. It is not clear what contracts. On the next line it says 'myocyte (astrocyte)' but probably means 'actinocyte' not the supporting cell of neurons in the brain. In the same section ciliary is used instead of flagella - noone considers sponge choanocytes to have cilia; it is not a question of semantics because it can be quite confusing to readers.

Neuropeptides is a term used to refer to a range of chemical signalling molecules. Neuropeptide means a very small molecule and should not be confused with other signalling molecules, so it would be better to say small molecules or chemical signalling molecules.

Genes/ molecules are said to be included (in the abstract) but the single line after sections stating that there is or isn't an expansion of genes is not helpful. On the contrary, the section addressing nerves almost only deals with molecules, not morphology. More care as to what data is included and not would improve the ability to arrive at the conclusion the author reaches.

5. Terminology of 'above', 'lower', 'below' are not useful in discussing phylogenetic relationships and these should be rephrased as sister to the remaining metazoa, or branched before or after a particular group. Similarly what is 'traditional' (line 25 page 1, also line 35-36), and what is 'usual' (pg 4 line 45-46)?

6. A time frame for the change in thinking is not described and might be useful for new readers. For example, how long has the Porifera first paradigm been in place, and what was it based on (examples of authors who concluded this and why)? Possibly a section just revisiting the arguments (as suggested earlier) including this paradigm would be a useful preface to the evaluation of the morphological data.

7. Combining ideas into sections: i) fossils, ii) theories: In addition to moving references to fossil data to one section on 'the fossil record (and its absence)' it would be useful to have a section on 'theories'. Under fossils would go the absence of a record for sponges lacking a skeleton (and what that might mean) and the new records for ctenophores (discussed under ctenophores) as well as the speculation of placozoan fossils. The concepts of steroid biomarkers would also fit here. Note that evidence for those as markers of demosponges is eroding with the finding of a strong sterol marker from Rhizaria (see *Nature Ecology & Evolution* 3(4) · March 2019). Under

the section on theories there could be an elaboration of the referred to theory in reference 19 (Nielsen's work) as well as theories currently in the section on Placozoa (regarding the plakula); elaboration of 'other theories' (pg 4 line 62) would also fit here. These sections would be much easier for newcomers to the field to quickly get the background for the problem.

Figure 1 shows characters but some are unclear (e.g. what are 'eumetazoan genes'), and neuropeptides A and B, which according to the descriptions in the text are small signalling molecules not neuropeptides. It is not immediately clear what this figure shows since the morphological characters are not well linked to transitions.

The discussion could use a greater argument building on synthesis of the data discussed in the previous sections. Synthesis and argument is lacking and so the conclusion lands abruptly without it being clear how it was arrived at. This may be a space consideration, but this is a thoughtful manuscript and very worth having if organized appropriately and with enough evaluation of the arguments to arrive at the conclusion stated.

Review form: Reviewer 2

Is the manuscript scientifically sound in its present form?

Yes

Are the interpretations and conclusions justified by the results?

Yes

Is the language acceptable?

Yes

Is it clear how to access all supporting data?

Not Applicable

Do you have any ethical concerns with this paper?

No

Have you any concerns about statistical analyses in this paper?

No

Recommendation?

Accept with minor revision (please list in comments)

Comments to the Author(s)

I enjoyed reading the manuscript "Early Animal Evolution: A morphologist's view" by Professor Claus Nielsen. The manuscript is a synthesis of our current knowledge of the early evolution of animals, with emphasis on evolutionary pathways possibly followed by morphological systems. The text is authoritative and sometimes speculative, but this is expected from this type of manuscript. The quality of the writing and content is up to the standards of previous work by Professor Nielsen. I would like to congratulate him for his contribution to the debate.

I have some suggestions, just for the sake of clarity and to make the text more accessible to readers who are not expert on this field.

-Page 1, line 26: "above", I understand the use of such terms is convenient, but they are not

precise. I would suggest rewording the sentence along the lines of sponges diverging first in the tree, ctenophores splitting later close to cnidarians.

-Page 1, line 41: "basal" is a term that it is losing support in the literature due to ambiguity. I would suggest replacing by "early diverging/splitting".

-Page 1, line 49: please, provide reference for filastereans as sister to animals.

-Page 1, line 53: "Unicellularity precludes differentiation into different cell types", I think this needs elaboration. All the lineages of non-animal holozoans display facultative multicellular stages with cell differentiation (Nicole King and Ruiz-Trillo work). And even during unicellular stages, they show sequential cell types segregated by time, not in space (idem). The gene systems used in those cell stages are most likely the same used to deploy different cell types in space and time in animals. Please, see recent reviews by Sebe-Pedros (Nature Rev Genetics 2017), Brunet and King (Development Cell 2018) and Paps (Integrative Comparative Biology 2018).

-Page 2, line 44: this sentence seems a bit out of place and could add more to the manuscript's argument. I would suggest fleshing it out, maybe mention that those new genes are related to animal multicellularity hallmarks (gene regulation, adhesion, cell cycle, etc). Those are all explained in the reviews cited in the previous point.

-Page 3, line 20: a recent paper has disputed the validity of Ediacaran sponge-markers, as these seem to be also found in Rhizaria (Nettersheim et al, Nature Ecology and Evolution 2019). This could be mentioned.

-Page 3, line 56: "A large number of genes found in the Eumetazoans are absent from the poriferans", this can also be said of ctenophores (Pisani et al PNAS 2015, Paps and Holland Nature Comms 2018, or Pett et al Molecular Biology and Evolution 2019).

-Page 4, line 19: I think that for the sake of non-experts, brief descriptions of cnidae and colloblast are needed.

-Page 4, line 46: similarly, a succinct explanation of what the 'usual' ultrastructure of synapses is (or at least put an example of animal).

-Page 5, lines 24-27: I think it is worth to mention that recent works with a significantly expanded placozoan sampling and using site-heterogeneous evolutionary models place placozoans as sister to cnidarians, please see Laumer et al eLife 2018, and Eitel et al PloS Biology 2018 (Supp Figs S15-S18).

-Page 6, line 11: the reference 93 on "animal" cholesterol found in Dickinsonia, the discussion of the very same paper acknowledges that all these molecules are also found in non-animal holozoans, calling into question their claim of the animal affiliation of Dickinsonia. They just decided to ignore it in the title of the paper.

-Page 6, line 40: the term "important molecules" requires clarification.

-Page 6, line 40: similarly to "basal", the expression "ancestral position of sponges" would need rewording.

-Figure 1: at the root of the tree, the idea of an obligate unicellular ancestor is problematic. As mentioned above, ichthyosporeans, filasterans, and choanoflagellates contain species with multicellular stages (some aggregative, some colonial).

-Figure 1: the figure does not include bilaterians, whose position is key to reconstruct some of the nodes. Or better said, the position of placozoans respect Cnidaria and Bilateria is essential.

Decision letter (RSOS-190638.R0)

08-May-2019

Dear Dr Nielsen,

The editors assigned to your paper ("Early Animal Evolution: A morphologist's view") have now received comments from reviewers. We would like you to revise your paper in accordance with

the referee and Associate Editor suggestions which can be found below (not including confidential reports to the Editor). Please note this decision does not guarantee eventual acceptance.

Please submit a copy of your revised paper before 31-May-2019. Please note that the revision deadline will expire at 00.00am on this date. If we do not hear from you within this time then it will be assumed that the paper has been withdrawn. In exceptional circumstances, extensions may be possible if agreed with the Editorial Office in advance. We do not allow multiple rounds of revision so we urge you to make every effort to fully address all of the comments at this stage. If deemed necessary by the Editors, your manuscript will be sent back to one or more of the original reviewers for assessment. If the original reviewers are not available, we may invite new reviewers.

- Data accessibility

<http://datadryad.org/submit?journalID=RSOS&manu=RSOS-190638>

- Competing interests

- Authors' contributions

- Acknowledgements

- Funding statement

on behalf of Dr David Ferrier (Associate Editor) and Kevin Padian (Subject Editor)
openscience@royalsociety.org

Associate Editor's comments (Dr David Ferrier):

Two expert referees have provided a number of helpful suggestions (and a third referee is due to provide their comments soon, which will be passed on if they arrive, but since these further comments are now overdue it seems unwise to delay sending the received comments any longer). Both referees find this manuscript to be of great interest and they provide comments that would strengthen the paper still further. All of their comments are aimed at making the manuscript as accessible as possible to the widest readership. It would clearly be desirable to incorporate as many of their suggestions as possible, even though some of these require a significant reworking of elements of the organisation of the current manuscript structure (hence the 'major revisions' decision).

One further comment on a matter close to my own heart is the citation of the recent Pastrana sponge ParaHox paper (reference [80]) on page 5. I would urge caution in the mention of this particular paper and would argue that describing it as casting "serious doubt" on the sponge Cdx

ParaHox gene classification as unwarranted. There are serious flaws in this Pastrana paper, in terms of weaknesses in their own analyses (lack of support values, lack of rooted phylogenies, use of unsuitable sequence evolution models, acceptance of a biologically unreasonable Anx classification) as well as their deliberate decision to ignore significant elements of the evidence (synteny) used in the previous literature that helped to classify the sponge gene as a Cdx. Whilst I agree that the paper can be cited as highlighting that the sponge Cdx story is still under debate, I think that it is not correct to say that the Pastrana paper casts serious doubt on the sponge Cdx classification. Further paper(s) on this topic should follow in the future!

Comments to Author:

Reviewers' Comments to Author:

Reviewer: 1

Comments to the Author(s)

Nielsen, Early Animal Evolution: A morphologist's view.

Nielsen reviews the morphological characters of each of the four non-bilaterian groups as well as choanoflagellates, with a view to evaluate evolutionary relationships.

The manuscript is a concise summary of the principal features of each group and as such could be not only extremely useful to a newcomer to the field, but also an excellent opinion piece, with a few organizational changes and some additions.

The abstract proposes an interesting hypothesis and although the text presents data, there is very limited analysis of the data, and a simple conclusion at the end saying the data supports the hypothesis is not easily digested. Below are some suggestions for changes that would improve this manuscript.

1. The manuscript sits between review and opinion piece. It would be more useful if it were more objective and if the text really evaluated character gain or loss in the two scenarios proposed, equally: by studying character gain and loss in the case Porifera branched first or in the case Ctenophora branched first. If that were added it would greatly improve the manuscript and increase interest.

2. The headers of subsections outline the author's argument, but given this is about morphology, it seems that the argument would be better placed in a section of its own at the beginning (using those headers) and then new headers might better be titles that highlight key characters that help determine, in a morphologist's view, the likelihood of moving from one state to another (following that argument). Header titles might be for example: 'Unicell to metazoan: cell differentiation' and 'Gaining an epithelium' or 'Epithelia and digestion: the gut'...something along those lines.

For example, choanoflagellates are described as single uniform unicells and the need for phagocytosis by each cell and lack of transfer of material between cells is a character that distinguishes them from metazoans. Some work suggest even similar cells can appear to have distinct characters in colonial flagellates (e.g. Laundon et al 2019), from a morphological view, but this is not yet supported by transcription of molecules. Whether colonies pass materials between one-another has not properly been addressed, and seems the point where Nielsen draws the line. However the text does not remind readers that these organisms are endpoints in evolutionary experiments and so some intermediate form that might have shared nutrients could have existed. Instead the argument is made that collared cells are innately similar, and by parsimony sponges arose from a collared ancestor. This is not necessarily the case however, since not all poriferans have collared cells - collared cells can be lost and gained even within the Porifera.

More emphasis should fall on examining the morphological transition from Porifera through other non-bilaterians and in turn, from Ctenophora through other non-bilaterians including Porifera. An examination of what losses must be considered were Porifera sister to Cnidaria and Placozoa would be helpful. A figure showing the gains on one scenario and losses on the other would be useful.

3. Key features of the different transitions are discussed (e.g. collagens) but not comprehensively.
 a) For example, Type IV collagen may not in fact be necessary for making epithelia. Even colonial filisterians can make good epithelia (see Dudin et al on *Sphaeroforma antarctica* in BioArchive), as can slime molds. What glues the cells together in metazoans (collagen) is not one of the characters Nielsen addresses, but seems like it might be a very useful character to evaluate in depth. It should at least be touched on.

b) Phagocytosis and digestion and transfer of food are discussed in one section but needs more attention. Contrary to what is said under the Choanoblastea section, digestion is quite well known in sponges (e.g. Willenz and Van de Vyver, Imsieke, Wilkinson). Can the present ctenophore groups tell us anything about that transition though? What about groups that may not have fossilized, just as Nielsen says that sponges without skeletons would not have fossilized. Absence of fossils means many groups with these characters could have existed. How a gut evolved is addressed under the gastrea section too. It would be useful if one section addressed feeding, digestion and the gut, with all groups.

c) Whether or not nerves arose and were lost is also addressed, but it seems cursory. This topic has been covered heavily by others but the relevance of the gain of nerves as a morphological character could be better discussed - a morphological view on this is lacking in the literature.

4. There are small but significant misunderstandings in these sections. For example, Pg 3 lines 8-9 refer to muscle - but a definition of muscle is needed. Presumably sponges have a type of smooth muscle (most authors from the earliest to recent) find this. It is not clear what contracts. On the next line it says 'myocyte (astrocyte)' but probably means 'actinocyte' not the supporting cell of neurons in the brain. In the same section ciliary is used instead of flagella - noone considers sponge choanocytes to have cilia; it is not a question of semantics because it can be quite confusing to readers.

Neuropeptides is a term used to refer to a range of chemical signalling molecules. Neuropeptide means a very small molecule and should not be confused with other signalling molecules, so it would be better to say small molecules or chemical signalling molecules.

Genes/molecules are said to be included (in the abstract) but the single line after sections stating that there is or isn't an expansion of genes is not helpful. On the contrary, the section addressing nerves almost only deals with molecules, not morphology. More care as to what data is included and not would improve the ability to arrive at the conclusion the author reaches.

5. Terminology of 'above', 'lower', 'below' are not useful in discussing phylogenetic relationships and these should be rephrased as sister to the remaining metazoa, or branched before or after a particular group. Similarly what is 'traditional' (line 25 page 1, also line 35-36), and what is 'usual' (pg 4 line 45-46)?

6. A time frame for the change in thinking is not described and might be useful for new readers. For example, how long has the Porifera first paradigm been in place, and what was it based on (examples of authors who concluded this and why)? Possibly a section just revisiting the

arguments (as suggested earlier) including this paradigm would be a useful preface to the evaluation of the morphological data.

7. Combining ideas into sections: i) fossils, ii) theories: In addition to moving references to fossil data to one section on 'the fossil record (and its absence)' it would be useful to have a section on 'theories'. Under fossils would go the absence of a record for sponges lacking a skeleton (and what that might mean) and the new records for ctenophores (discussed under ctenophores) as well as the speculation of placozoan fossils. The concepts of steroid biomarkers would also fit here. Note that evidence for those as markers of demosponges is eroding with the finding of a strong sterol marker from Rhizaria (see *Nature Ecology & Evolution* 3(4) · March 2019). Under the section on theories there could be an elaboration of the referred to theory in reference 19 (Nielsen's work) as well as theories currently in the section on Placozoa (regarding the plakula); elaboration of 'other theories' (pg 4 line 62) would also fit here. These sections would be much easier for newcomers to the field to quickly get the background for the problem.

Figure 1 shows characters but some are unclear (e.g. what are 'eumetazoan genes'), and neuropeptides A and B, which according to the descriptions in the text are small signalling molecules not neuropeptides. It is not immediately clear what this figure shows since the morphological characters are not well linked to transitions.

The discussion could use a greater argument building on synthesis of the data discussed in the previous sections. Synthesis and argument is lacking and so the conclusion lands abruptly without it being clear how it was arrived at. This may be a space consideration, but this is a thoughtful manuscript and very worth having if organized appropriately and with enough evaluation of the arguments to arrive at the conclusion stated.

Reviewer: 2

Comments to the Author(s)

I enjoyed reading the manuscript "Early Animal Evolution: A morphologist's view" by Professor Claus Nielsen. The manuscript is a synthesis of our current knowledge of the early evolution of animals, with emphasis on evolutionary pathways possibly followed by morphological systems. The text is authoritative and sometimes speculative, but this is expected from this type of manuscript. The quality of the writing and content is up to the standards of previous work by Professor Nielsen. I would like to congratulate him for his contribution to the debate.

I have some suggestions, just for the sake of clarity and to make the text more accessible to readers who are not expert on this field.

-Page 1, line 26: "above", I understand the use of such terms is convenient, but they are not precise. I would suggest rewording the sentence along the lines of sponges diverging first in the tree, ctenophores splitting later close to cnidarians.

-Page 1, line 41: "basal" is a term that it is losing support in the literature due to ambiguity. I would suggest replacing by "early diverging/splitting".

-Page 1, line 49: please, provide reference for filastereans as sister to animals.

-Page 1, line 53: "Unicellularity precludes differentiation into different cell types", I think this needs elaboration. All the lineages of non-animal holozoans display facultative multicellular stages with cell differentiation (Nicole King and Ruiz-Trillo work). And even during unicellular stages, they show sequential cell types segregated by time, not in space (idem). The gene systems used in those cell stages are most likely the same used to deploy different cell types in space and time in animals. Please, see recent reviews by Sebe-Pedros (*Nature Rev Genetics* 2017), Brunet and King (*Development Cell* 2018) and Paps (*Integrative Comparative Biology* 2018).

-Page 2, line 44: this sentence seems a bit out of place and could add more to the manuscript's argument. I would suggest fleshing it out, maybe mention that those new genes are related to animal multicellularity hallmarks (gene regulation, adhesion, cell cycle, etc). Those are all explained in the reviews cited in the previous point.

-Page 3, line 20: a recent paper has disputed the validity of Ediacaran sponge-markers, as these seem to be also found in Rhizaria (Nettersheim et al, Nature Ecology and Evolution 2019). This could be mentioned.

-Page 3, line 56: "A large number of genes found in the Eumetazoans are absent from the poriferans", this can also be said of ctenophores (Pisani et al PNAS 2015, Paps and Holland Nature Comms 2018, or Pett et al Molecular Biology and Evolution 2019).

-Page 4, line 19: I think that for the sake of non-experts, brief descriptions of cnidae and colloblast are needed.

-Page 4, line 46: similarly, a succinct explanation of what the 'usual' ultrastructure of synapses is (or at least put an example of animal).

-Page 5, lines 24-27: I think it is worth to mention that recent works with a significantly expanded placozoan sampling and using site-heterogeneous evolutionary models place placozoans as sister to cnidarians, please see Laumer et al eLife 2018, and Eitel et al PloS Biology 2018 (Supp Figs S15-S18).

-Page 6, line 11: the reference 93 on "animal" cholesterol found in Dickinsonia, the discussion of the very same paper acknowledges that all these molecules are also found in non-animal holozoans, calling into question their claim of the animal affiliation of Dickinsonia. They just decided to ignore it in the title of the paper.

-Page 6, line 40: the term "important molecules" requires clarification.

-Page 6, line 40: similarly to "basal", the expression "ancestral position of sponges" would need rewording.

-Figure 1: at the root of the tree, the idea of an obligate unicellular ancestor is problematic. As mentioned above, ichthyosporeans, filasterans, and choanoflagellates contain species with multicellular stages (some aggregative, some colonial).

-Figure 1: the figure does not include bilaterians, whose position is key to reconstruct some of the nodes. Or better said, the position of placozoans respect Cnidaria and Bilateria is essential.

Author's Response to Decision Letter for (RSOS-190638.R0)

See Appendices A & B.

RSOS-190638.R1 (Revision)

Review form: Reviewer 1

Is the manuscript scientifically sound in its present form?

Yes

Are the interpretations and conclusions justified by the results?

Yes

Is the language acceptable?

Yes

Is it clear how to access all supporting data?

Not Applicable

Do you have any ethical concerns with this paper?

No

Have you any concerns about statistical analyses in this paper?

No

Recommendation?

Accept with minor revision (please list in comments)

Comments to the Author(s)

Review of Nielsen – Open Science

This revision reads very well. I found a few small errors and have a number of small comments which I list below, with tables and figures first.

Table 1:

Spelling of Nidogen (Nodogen)

Cells with a collar complex – the text refers to these in all metazoans, and yet here in the table they are only listed in Porifera and unicells. The text refers to choanocytes and collar cells as two different things – perhaps best to either remove this or rename these here to be consistent with the text.

‘metazoan genes’ and ‘eumetazoan genes’ – needs reference to the text or some indication of what these are.

Figure 3 caption: typo ‘based on losses’

I think the new figures are excellent additions.

Manuscript text:

Abstract:

L 24 typo: Important

L 29 ‘Ctenophore-first’ theory (keep the same terminology throughout)

Introduction:

P2/23 Line 58 – cannot say ‘basal animal group’ here. Change this to sister group to all other metazoan, as is correctly written on line 39 and later in the subheading below on line 56.

P3/23 Line 13 – what is a ‘basal’ vane? Is that the supposed location of the vane on the flagellum? I think it runs top to bottom.

L 15 – what is ‘continuity’ between the cells?

P3/23 L48 – ‘show their evolution from’ – assumes homology, but it’s an assumption of the author. There is no evidence these structures are either homologous or convergent and so it should be left open question.

L58 - 'lower' is also not generally accepted now unless referring specifically to the physical position on a picture of a tree. Use 'non-bilaterian' instead, which would work well here.

P4/23 Line 5 - note that there is also no basement membrane in many (most?) acoels, which I find pretty interesting! I haven't looked for perlecan or nidogen there though.

L15 - reword to change 'is' to 'are' for ease of reading: suggest 'There are different theories...'

L22 - completely agree.

L38-39 - loss: is this sentence needed? Or if kept then elaborate to say that the electrical signal travels through the entire animal's syncytial tissues (not just a choanosyncytium...which is part of the whole syncytium).

L41-42 - There's a sentence that seems incomplete and possibly left in. Or a period is missing. In the responses document the author says all fossil references are removed, but this and later reference to Dickinsonia are left in.

P5/23 Line 12-13 - remove 'for'

L14 - Diversity of 'what'? Of morphologies? This probably doesn't refer to species diversity?

L40 - 'The cnidarians' - maybe clarify this is the larva.

L43 - 'Ctenophore' - cydippid or adult? These two sentences are comparing structures on larvae with those on adults - some clarification is needed about what is directly comparable especially since the ctenophore larva is really a miniature adult (the larva that we know today), while the cnidarian larva comes in a range of morphologies, and none are like the adult.

P6/23 - Note that the epitheliomuscular cells are also striated.

L34 - There is actually no 5HT in cnidarians (melanopsin receptors are the closest) - see Anctil 2009 (Anctil M. 2009. Chemical transmission in the sea anemone *Nematostella vectensis*: A genomic perspective. *Comparative Biochemistry and Physiology Part D: Genomics and Proteomics* 4(4):268-289.) And also Bosch et al 2017. Bosch TCG, Klimovich A, Domazet-Lošo T, Gründer S, Holstein TW, Jékely G, Miller DJ, Murillo-Rincon AP, Rentzsch F, Richards GS et al. . 2017. Back to the basics: Cnidarians start to fire. *Trends in Neurosciences* 40(2):92-105. There is no 5HT in ctenophores (see Moroz 2015 Moroz LL. 2015. Convergent evolution of neural systems in ctenophores. *The Journal of Experimental Biology* 218(4):598-611. And supplemental data in Moroz et al 2014.).

P8/23 - last line - what is meant by 'unsupported groups'?

References:

3 - Parentheses are around a blank (no Doi?)

10 - this reference should be 2019, MBE (it is now published)

23 - Lauden et al needs the Doi or some location information.

42 - 'eds' is written twice

53 - 'eds' is also written twice

Review form: Reviewer 2

Is the manuscript scientifically sound in its present form?

Yes

Are the interpretations and conclusions justified by the results?

Yes

Is the language acceptable?

Yes

Is it clear how to access all supporting data?

Not Applicable

Do you have any ethical concerns with this paper?

No

Have you any concerns about statistical analyses in this paper?

No

Recommendation?

Accept as is

Comments to the Author(s)

I want to congratulate the author for the efforts made. Although we will have to agree to disagree on the multicellularity in non-metazoan holozoans!

I think the manuscript has improved much and it is now ready for publication.

Decision letter (RSOS-190638.R1)

03-Jul-2019

Dear Dr Nielsen:

On behalf of the Editors, I am pleased to inform you that your Manuscript RSOS-190638.R1 entitled "Early Animal Evolution: A morphologist's view" has been accepted for publication in Royal Society Open Science subject to minor revision in accordance with the referee suggestions. Please find the referees' comments at the end of this email.

The reviewers and Subject Editor have recommended publication, but also suggest some minor revisions to your manuscript. Therefore, I invite you to respond to the comments and revise your manuscript.

- Ethics statement

- Data accessibility

If you wish to submit your supporting data or code to Dryad (<http://datadryad.org/>), or modify your current submission to dryad, please use the following link:
<http://datadryad.org/submit?journalID=RSOS&manu=RSOS-190638.R1>

- Competing interests

- Authors' contributions

- Acknowledgements

- Funding statement

Because the schedule for publication is very tight, it is a condition of publication that you submit the revised version of your manuscript before 12-Jul-2019. Please note that the revision deadline will expire at 00.00am on this date. If you do not think you will be able to meet this date please let me know immediately.

on behalf of Dr David Ferrier (Associate Editor) and Kevin Padian (Subject Editor)
openscience@royalsociety.org

Associate Editor Comments to Author (Dr David Ferrier):

Both referees are appreciative of the efforts made to address their comments and

recommendations, thank you. One referee has spotted a number of further minor changes that should be relatively easy to incorporate in order to make this manuscript acceptable for publication.

Subject Editor Comments to Author:

Thanks very much for your contribution and I hope the few remaining changes will be easy to make.

Reviewer comments to Author:

Reviewer: 2

Comments to the Author(s)

I want to congratulate the author for the efforts made. Although we will have to agree to disagree on the multicellularity in non-metazoan holozoans!

I think the manuscript has improved much and it is now ready for publication.

Reviewer: 1

Comments to the Author(s)

Review of Nielsen – Open Science

This revision reads very well. I found a few small errors and have a number of small comments which I list below, with tables and figures first.

Table 1:

Spelling of Nidogen (Nodogen)

Cells with a collar complex – the text refers to these in all metazoans, and yet here in the table they are only listed in Porifera and unicells. The text refers to choanocytes and collar cells as two different things – perhaps best to either remove this or rename these here to be consistent with the text.

‘metazoan genes’ and ‘eumetazoan genes’ – needs reference to the text or some indication of what these are.

Figure 3 caption: typo ‘based on losses’

I think the new figures are excellent additions.

Manuscript text:

Abstract:

L 24 typo: Important

L 29 ‘Ctenophore-first’ theory (keep the same terminology throughout)

Introduction:

P2/23 Line 58 – cannot say ‘basal animal group’ here. Change this to sister group to all other metazoan, as is correctly written on line 39 and later in the subheading below on line 56.

P3/23 Line 13 – what is a ‘basal’ vane? Is that the supposed location of the vane on the flagellum? I think it runs top to bottom.

L 15 – what is ‘continuity’ between the cells?

P3/23 L48 - 'show their evolution from' - assumes homology, but it's an assumption of the author. There is no evidence these structures are either homologous or convergent and so it should be left open question.

L58 - 'lower' is also not generally accepted now unless referring specifically to the physical position on a picture of a tree. Use 'non-bilaterian' instead, which would work well here.

P4/23 Line 5 - note that there is also no basement membrane in many (most?) acoels, which I find pretty interesting! I haven't looked for perlecan or nidogen there though.

L15 - reword to change 'is' to 'are' for ease of reading: suggest 'There are different theories...'

L22 - completely agree.

L38-39 - loss: is this sentence needed? Or if kept then elaborate to say that the electrical signal travels through the entire animal's syncytial tissues (not just a choanosyncytium...which is part of the whole syncytium).

L41-42 - There's a sentence that seems incomplete and possibly left in. Or a period is missing. In the responses document the author says all fossil references are removed, but this and later reference to Dickinsonia are left in.

P5/23 Line 12-13 - remove 'for'

L14 - Diversity of 'what'? Of morphologies? This probably doesn't refer to species diversity?

L40 - 'The cnidarians' - maybe clarify this is the larva.

L43 - 'Ctenophore' - cydippid or adult? These two sentences are comparing structures on larvae with those on adults - some clarification is needed about what is directly comparable especially since the ctenophore larva is really a miniature adult (the larva that we know today), while the cnidarian larva comes in a range of morphologies, and none are like the adult.

P6/23 - Note that the epitheliomuscular cells are also striated.

L34 - There is actually no 5HT in cnidarians (melanopsin receptors are the closest) - see Anctil 2009 (Anctil M. 2009. Chemical transmission in the sea anemone *Nematostella vectensis*: A genomic perspective. *Comparative Biochemistry and Physiology Part D: Genomics and Proteomics* 4(4):268-289.) And also Bosch et al 2017. Bosch TCG, Klimovich A, Domazet-Lošo T, Gründer S, Holstein TW, Jékely G, Miller DJ, Murillo-Rincon AP, Rentzsch F, Richards GS et al. . 2017. Back to the basics: Cnidarians start to fire. *Trends in Neurosciences* 40(2):92-105. There is no 5HT in ctenophores (see Moroz 2015 Moroz LL. 2015. Convergent evolution of neural systems in ctenophores. *The Journal of Experimental Biology* 218(4):598-611. And supplemental data in Moroz et al 2014.).

P8/23 - last line - what is meant by 'unsupported groups'?

References:

3 - Parentheses are around a blank (no Doi?)

10 - this reference should be 2019, MBE (it is now published)

23 - Lauden et al needs the Doi or some location information.

42 - 'eds' is written twice

53 - 'eds' is also written twice

Author's Response to Decision Letter for (RSOS-190638.R1)

See Appendix C.

Decision letter (RSOS-190638.R2)

04-Jul-2019

Dear Dr Nielsen,

I am pleased to inform you that your manuscript entitled "Early Animal Evolution: A morphologist's view" is now accepted for publication in Royal Society Open Science.

on behalf of Dr David Ferrier (Associate Editor) and Kevin Padian (Subject Editor)
openscience@royalsociety.org

Appendix A

Reviewer: 1

Comments to the Author(s)

Nielsen, Early Animal Evolution: A morphologist's view.

Nielsen reviews the morphological characters of each of the four non-bilaterian groups as well as choanoflagellates, with a view to evaluate evolutionary relationships.

The manuscript is a concise summary of the principal features of each group and as such could be not only extremely useful to a newcomer to the field, but also an excellent opinion piece, with a few organizational changes and some additions.

My comments are indicated by • xxx •

•Abstract and Discussion have been rewritten •

The abstract proposes an interesting hypothesis and although the text presents data, there is very limited analysis of the data, and a simple conclusion at the end saying the data supports the hypothesis is not easily digested. Below are some suggestions for changes that would improve this manuscript. • Analyses of all the datasets would make the manuscript enormous, and analyses of the more molecular-based data are beyond my capacity. I have added two new illustrations to the Discussion and highlighted the consequences inherent in the Ctenophora-first hypothesis. This is added to the Abstract and the Discussion has been expanded •

1. The manuscript sits between review and opinion piece. It would be more useful if it were more objective and if the text really evaluated character gain or loss in the two scenarios proposed, equally: by studying character gain and loss in the case Porifera branched first or in the case Ctenophora branched first. If that were added it would greatly improve the manuscript and increase interest • see above •

2. The headers of subsections outline the author's argument, but given this is about morphology, it seems that the argument would be better placed in a section of its own at the beginning (using those headers) and then new headers might better be titles that highlight key characters that help determine, in a morphologist's view, the likelihood of moving from one state to another (following that argument). Header titles might be for example: 'Unicell to metazoan: cell differentiation' and 'Gaining an epithelium' or 'Epithelia and digestion: the gut'...something along those lines.

• I have thought a lot about your suggestion, and it might work for the first steps of the evolution (the origin of multicellularity), but as far as I can see, it will become quite confusing at the next levels, so I have stayed with my original organization of the text •

For example, choanoflagellates are described as single uniform unicells and the need for phagocytosis by each cell and lack of transfer of material between cells is a character that distinguishes them from metazoans. Some work suggest even similar cells can appear to have distinct characters in colonial flagellates (e.g. Laundon et al 2019), from a morphological view, but this is not yet supported by transcription of molecules. • I have changed the text to specify that non-feeding cells do not occur in the unicellular organisms. • Whether colonies pass materials between one-another has not properly been addressed, and seems the point where Nielsen draws the line. However the text does not remind readers that these organisms are endpoints in evolutionary experiments and so some intermediate form that might have shared nutrients could have existed. • It can of course be imagined, but is there any hint of such a case? • Instead the argument is made that collared cells are innately similar, and by parsimony sponges arose from a collared ancestor. This is not necessarily the case however, since not all poriferans have collared cells - collared cells can be lost and gained even within the Porifera. • Information about the peculiar carnivorous sponges has been added. I know of no example of 'gain of collared cells' in Porifera •

More emphasis should fall on examining the morphological transition from Porifera through other non-bilaterians and in turn, from Ctenophora through other non-bilaterians including Porifera. An examination of what losses must be considered were Porifera sister to Cnidaria and Placozoa would be helpful. A figure showing the gains on one scenario and losses on the other would be useful. • Two new diagrams have been added to the discussion •

3. Key features of the different transitions are discussed (e.g. collagens) but not comprehensively.

a) For example, Type IV collagen may not in fact be necessary for making epithelia. Even colonial filisterians can make good epithelia (see Dudin et al on *Sphaeroforma antarctica* in BioArchive), as can slime molds. What glues the cells together in metazoans (collagen) is not one of the characters Nielsen addresses, but seems like it might be a very useful character to evaluate in depth. It should at least be touched on. • text changed •

b) Phagocytosis and digestion and transfer of food are discussed in one section but needs more attention. Contrary to what is said under the Choanoblastea section, digestion is quite well known in sponges (e.g. Willenz and Van de Vyver, Imsieke, Wilkinson). • information added • Can the present ctenophore groups tell us anything about that transition though? What about groups that may not have fossilized, just as Nielsen says that sponges without skeletons would not have fossilized. Absence of fossils means many groups with these characters could have existed. How a gut evolved is addressed under the gastrea section too. It would be useful if one section addressed feeding, digestion and the gut, with all groups. • It would indeed be interesting to compare digestion in the main clades, but as far as I know, nothing is known about the ctenophores. The gain/loss of a gut in the two scenarios are discussed together with the two new diagrams •

c) Whether or not nerves arose and were lost is also addressed, but it seems cursory. This topic has been covered heavily by others but the relevance of the gain of nerves as a morphological character could be better discussed - a morphological view on this is lacking in the literature. • new sentence added •

4. There are small but significant misunderstandings in these sections. For example, Pg 3 lines 8-9 refer to muscle - but a definition of muscle is needed. Presumably sponges have a type of smooth muscle (most authors from the earliest to recent) find this. It is not clear what contracts. On the next line it says 'myocyte (astrocyte)' but probably means 'actinocyte' not the supporting cell of neurons in the brain • corrected • In the same section ciliary is used instead of flagella - noone considers sponge choanocytes to have cilia; it is not a question of semantics because it can be quite confusing to readers. • I usually talk about two types of cilia: undulatory cilia (usually called flagella – different from the bacterial flagella) and effective-stroke cilia. I prefer to keep to my usual nomenclature •

Neuropeptides is a term used to refer to a range of chemical signalling molecules. Neuropeptide means a very small molecule and should not be confused with other signalling molecules, so it would be better to say small molecules or chemical signalling molecules. • changed to neurotransmitters •

Genes/molecules are said to be included (in the abstract • deleted •) but the single line after sections stating that there is or isn't an expansion of genes is not helpful. On the contrary, the section addressing nerves almost only deals with molecules, not morphology. More care as to what data is included and not would improve the ability to arrive at the conclusion the author reaches. • sentence added •

5. Terminology of 'above', 'lower', 'below' are not useful in discussing phylogenetic relationships and these should be rephrased as sister to the remaining metazoa, or branched before or after a particular group.

Similarly what is 'traditional' (line 25 page 1, also line 35-36), and what is 'usual' (pg 4 line 45-46)? • changed •

6. A time frame for the change in thinking is not described and might be useful for new readers. For example, how long has the Porifera first paradigm been in place, and what was it based on (examples of authors who concluded this and why)? Possibly a section just revisiting the arguments (as suggested earlier) including this paradigm would be a useful preface to the evaluation of the morphological data. • Sentence added at the beginning of the text •

7. Combining ideas into sections: i) fossils, ii) theories: In addition to moving references to fossil data to one section on 'the fossil record (and its absence)' it would be useful to have a section on 'theories'. Under fossils would go the absence of a record for sponges lacking a skeleton (and what that might mean) and the new records for ctenophores (discussed under ctenophores) as well as the speculation of placozoan fossils. • information on fossils has generally been removed • The concepts of steroid biomarkers would also fit here. Note that evidence for those as markers of demosponges is eroding with the finding of a strong sterol marker from Rhizaria (see Nature Ecology & Evolution 3(4) · March 2019). Under the section on theories there could be an elaboration of the referred to theory in reference 19 (Nielsen's work) as well as theories currently in the section on Placozoa (regarding the plakula); elaboration of 'other theories' (pg 4 line 62) would also fit here. These sections would be much easier for newcomers to the field to quickly get the background for the problem.

Figure 1 shows characters but some are unclear (e.g. what are 'eumetazoan genes'), and neuropeptides A and B, which according to the descriptions in the text are small signalling molecules not neuropeptides • changed to neurotransmitters •. It is not immediately clear what this figure shows since the morphological characters are not well linked to transitions.

The discussion could use a greater argument building on synthesis of the data discussed in the previous sections. Synthesis and argument is lacking and so the conclusion lands abruptly without it being clear how it was arrived at. This may be a space consideration, but this is a thoughtful manuscript and very worth having if organized appropriately and with enough evaluation of the arguments to arrive at the conclusion stated. • New illustrations added and discussion expanded •

Appendix B

Reviewer: 2

Comments to the Author(s)

I enjoyed reading the manuscript “Early Animal Evolution: A morphologist's view” by Professor Claus Nielsen. The manuscript is a synthesis of our current knowledge of the early evolution of animals, with emphasis on evolutionary pathways possibly followed by morphological systems. The text is authoritative and sometimes speculative, but this is expected from this type of manuscript. The quality of the writing and content is up to the standards of previous work by Professor Nielsen. I would like to congratulate him for his contribution to the debate.

I have some suggestions, just for the sake of clarity and to make the text more accessible to readers who are not expert on this field.

•Abstract and discussion have been rewritten•

-Page 1, line 26: “above”, I understand the use of such terms is convenient, but they are not precise. I would suggest rewording the sentence along the lines of sponges diverging first in the tree, ctenophores splitting later close to cnidarians. • changed •

-Page 1, line 41: “basal” is a term that it is losing support in the literature due to ambiguity. I would suggest replacing by “early diverging/splitting”. • changed •

-Page 1, line 49: please, provide reference for filastereans as sister to animals. • information about ichthyosporeans and filastreans added •

-Page 1, line 53: “Unicellularity precludes differentiation into different cell types”, I think this needs elaboration. All the lineages of non-animal holozoans display facultative multicellular stages with cell differentiation (Nicole King and Ruiz-Trillo work). And even during unicellular stages, they show sequential cell types segregated by time, not in space (idem). The gene systems used in those cell stages are most likely the same used to deploy different cell types in space and time in animals. Please, see recent reviews by Sebe-Pedros (Nature Rev Genetics 2017), Brunet and King (Development Cell 2018) and Paps (Integrative Comparative Biology 2018). • As far as I can see, there are no reports of multicellularity in these organisms. They are all colonial or multinucleate •

-Page 2, line 44: this sentence seems a bit out of place and could add more to the manuscript’s argument. I would suggest fleshing it out, maybe mention that those new genes are related to animal multicellularity hallmarks (gene regulation, adhesion, cell cycle, etc). Those are all explained in the reviews cited in the previous point. • modified and examples added •

-Page 3, line 20: a recent paper has disputed the validity of Ediacaran sponge-markers, as these seem to be also found in Rhizaria (Nettersheim et al, Nature Ecology and Evolution 2019). This could be mentioned. • deleted •

-Page 3, line 56: “A large number of genes found in the Eumetazoans are absent from the poriferans”, this can also be said of ctenophores (Pisani et al PNAS 2015, Paps and Holland Nature Comms 2018, or Pett et al Molecular Biology and Evolution 2019). • reference added •

-Page 4, line 19: I think that for the sake of non-experts, brief descriptions of cnidae and colloblast are needed. • explanations added •

-Page 4, line 46: similarly, a succinct explanation of what the ‘usual’ ultrastructure of synapses is (or at least put an example of animal). • added •

-Page 5, lines 24-27: I think it is worth to mention that recent works with a significantly expanded placozoan sampling and using site-heterogeneous evolutionary models place placozoans as sister to cnidarians, please see Laumer et al eLife 2018 • added •, and Eitel et al PloS Biology 2018 (Supp Figs S15-S18) • ??? •.

-Page 6, line 11: the reference 93 on “animal” cholesterol found in Dickinsonia, the discussion of the very same paper acknowledges that all these molecules are also found in non-animal holozoans, calling into

question their claim of the animal affiliation of Dickinsonia. They just decided to ignore it in the title of the paper. • second reference deleted •

- Page 6, line 40: the term “important molecules” requires clarification.
- Page 6, line 40: similarly to “basal”, the expression “ancestral position of sponges” would need rewording.
- changed – the conclusion has been expanded and new figures added •
- Figure 1: at the root of the tree, the idea of an obligate unicellular ancestor is problematic. As mentioned above, ichthyosporeans, filasterans, and choanoflagellates contain species with multicellular stages (some aggregative, some colonial). • again, these types are not multicellular •
- Figure 1: the figure does not include bilaterians, whose position is key to reconstruct some of the nodes. Or better said, the position of placozoans respect Cnidaria and Bilateria is essential. • It is mentioned in the Introduction, that the Cnidarians are used to represent the ‘Planulozoa or Gastraeozoa’, and the monophyly of the ‘Parahoxozoa’ seems very well established. I think that details about bilaterians will not change the main discussion, which is the question about the position of the ctenophores •

Appendix C

RSOS

Dear Editors,

Many thanks for the very useful comments, which have been easy to deal with. There were no specific comments from Reviewer 2, and my comments to Reviewer 1 are found below.

I hope the manuscript can now be accepted.

Very best wishes. Yours sincerely, Claus Nielsen

Reviewer: 1

Comments to the Author(s)

Review of Nielsen – Open Science

This revision reads very well. I found a few small errors and have a number of small comments, which I list below, with tables and figures first.

Table 1:

Spelling of Nidogen (Nodogen) •OK•

Cells with a collar complex – the text refers to these in all metazoans, and yet here in the table they are only listed in Porifera and unicells. The text refers to choanocytes and collar cells as two different things – perhaps best to either remove this or rename these here to be consistent with the text. •The terminology is difficult. Choanoflagellates and choanocytes have collar complexes which are used in particle collection. Collar cells of various types have a ring of usually short microvilli and a usually short cilium, but they don't collect particles. I think that it will only add to the confusion if I try to change the terminology. I have added 'Collar complexes consist of a ring of microvilli surrounding an undulating cilium and function in water transport and particle collection' to the legend •

'metazoan genes' and 'eumetazoan genes' – needs reference to the text or some indication of what these are. • The reference to Srivastava et al. gives the details •

Figure 3 caption: typo 'based on losses" •OK •

I think the new figures are excellent additions.

Manuscript text:

Abstract:

L 24 typo: Important •OK •

L 29 'Ctenophore-first' theory (keep the same terminology throughout) • corrected •

Introduction:

P2/23 Line 58 – cannot say 'basal animal group' here. Change this to sister group to all other metazoan, as is correctly written on line 39 and later in the subheading below on line 56. • changed •

P3/23 Line 13 – what is a 'basal' vane? Is that the supposed location of the vane on the flagellum? I think it runs top to bottom. • basal deleted •

L 15 – what is 'continuity' between the cells? • changed to no exchange of nutrients between the cells •

P3/23 L48 – ‘show their evolution from’ – assumes homology, but it’s an assumption of the author. There is no evidence these structures are either homologous or convergent and so it should be left open question. • The assumption is that of Arendt et al ref 24. •

L58 – ‘lower’ is also not generally accepted now unless referring specifically to the physical position on a picture of a tree. Use ‘non-bilaterian’ instead, which would work well here. • changed •

P4/23 Line 5 – note that there is also no basement membrane in many (most?) acoels, which I find pretty •interesting! I haven’t looked for perlecan or nidogen there though.

L15 – reword to change ‘is’ to ‘are’ for ease of reading: suggest ‘There are different theories...’ • changed •

L22 – completely agree.

L38-39 – loss: is this sentence needed? Or if kept then elaborate to say that the electrical signal travels through the entire animal’s syncytial tissues (not just a choanosyncytium...which is part of the whole syncytium). • changed to syncytial tissues •

L41-42 – There’s a sentence that seems incomplete and possibly left in. Or a period is missing. In the responses document the author says all fossil references are removed, but this and later reference to Dickinsonia are left in. • period added. A few references to fossils have been kept •

P5/23 Line 12-13 – remove ‘for’ • OK •

L14 – Diversity of ‘what’? Of morphologies? This probably doesn’t refer to species diversity? • morphological added •

L40 – ‘The cnidarians’ – maybe clarify this is the larva. • done •

L43 – ‘Ctenophore’ – cydippid or adult? These two sentences are comparing structures on larvae with those on adults – some clarification is needed about what is directly comparable especially since the ctenophore larva is really a miniature adult (the larva that we know today), while the cnidarian larva comes in a range of morphologies, and none are like the adult. • explained •

P6/23 – Note that the epitheliomuscular cells are also striated. • no action •

L34 – There is actually no 5HT in cnidarians (melanopsin receptors are the closest) – see Anctil 2009 (Anctil M. 2009. Chemical transmission in the sea anemone *Nematostella vectensis*: A genomic perspective. *Comparative Biochemistry and Physiology Part D: Genomics and Proteomics* 4(4):268-289.) And also Bosch et al 2017. Bosch TCG, Klimovich A, Domazet-Lošo T, Gründer S, Holstein TW, Jékely G, Miller DJ, Murillo-Rincon AP, Rentzsch F, Richards GS et al. . 2017. Back to the basics: Cnidarians start to fire. *Trends in Neurosciences* 40(2):92-105.

There is no 5HT in ctenophores (see Moroz 2015 Moroz LL. 2015. Convergent evolution of neural systems in ctenophores. *The Journal of Experimental Biology* 218(4):598-611. And supplemental data in Moroz et al 2014.). • 5HT removed from the list •

P8/23 – last line – what is meant by ‘unsupported groups’? • changed to groups with no synapomorphies •

References: • OK •

3 – Parentheses are around a blank (no Doi?)

10 – this reference should be 2019, MBE (it is now published)

23 – Lauden et al needs the Doi or some location information.

42 – ‘eds’ is written twice

53 – ‘eds’ is also written twice